# Plant and Arthropod IgE-Binding Papain-like Cysteine Proteases: Multiple Contributions to Allergenicity

**DOI:** 10.3390/foods13050790

**Published:** 2024-03-04

**Authors:** Ivana Giangrieco, Maria Antonietta Ciardiello, Maurizio Tamburrini, Lisa Tuppo, Adriano Mari, Claudia Alessandri

**Affiliations:** 1Institute of Biosciences and BioResources (IBBR), National Research Council of Italy (CNR), 80131 Naples, Italy; ivana.giangrieco@ibbr.cnr.it (I.G.); maurizio.tamburrini@ibbr.cnr.it (M.T.); lisa.tuppo@ibbr.cnr.it (L.T.); 2Associated Centers for Molecular Allergology (CAAM), 00100 Rome, Italy; mariaantonietta.ciardiello@ibbr.cnr.it (M.A.C.); adriano.mari@caam-allergy.com (A.M.); 3Allergy Data Laboratories (ADL), 04100 Latina, Italy

**Keywords:** sensitisation, tissue permeabilisation, plant food, mite proteases, tight junction, gastrointestinal digestion, inhaled allergens, allergen homologs, occupational allergy, proteolytic activity

## Abstract

Papain-like cysteine proteases are widespread and can be detected in all domains of life. They share structural and enzymatic properties with the group’s namesake member, papain. They show a broad range of protein substrates and are involved in several biological processes. These proteases are widely exploited for food, pharmaceutical, chemical and cosmetic biotechnological applications. However, some of them are known to cause allergic reactions. In this context, the objective of this review is to report an overview of some general properties of papain-like cysteine proteases and to highlight their contributions to allergy reactions observed in humans. For instance, the literature shows that their proteolytic activity can cause an increase in tissue permeability, which favours the crossing of allergens through the skin, intestinal and respiratory barriers. The observation that allergy to PLCPs is mostly detected for inhaled proteins is in line with the reports describing mite homologs, such as Der p 1 and Der f 1, as major allergens showing a frequent correlation between sensitisation and clinical allergic reactions. In contrast, the plant food homologs are often digested in the gastrointestinal tract. Therefore, they only rarely can cause allergic reactions in humans. Accordingly, they are reported mainly as a cause of occupational diseases.

## 1. Introduction

A proteolytic enzyme is a protein, which cleaves the peptide bond between two amino acids in a peptide or protein [1]. Proteases that have arisen from a single evolutionary origin are grouped in a clan [2]. Each clan is identified with two letters, the first representing the catalytic type of the protein families included in it [3]. Papain-like cysteine proteases (PLCPs) share structural and enzymatic properties with the group’s namesake member, papain. These proteases belong to the cysteine peptidase family C1, sub-family C1A (papain family, clan CA) [4]. Family C1 contains many endopeptidases and a few exopeptidases [3]. Papain is the best-known cysteine protease, which was isolated as early as 1879. As a classical cysteine protease, it has been widely used in the food, pharmaceutical, chemical and cosmetic fields [5]. For similar industrial applications other plant food components of the same protein family have also been exploited. Figure 1 schematizes some possible applications of actinidin from kiwifruit, papain and chymopapain from papaya fruit and latex, bromelain from pineapple and ficin from fig fruit and latex.

This figure also highlights that the exposure to these proteases can occur not only through the consumption of the plant food sources. In fact, PLCPs can be present, sometimes unexpectedly, in several other products, such as meat, cheese, bread, textile tissues and drugs. This occurrence might represent a risk for some people suffering from pathological reactions to these proteases because they are exposed to hidden molecules. However, food PLCPs can also be inhaled, due to exposure to airborne particles. Therefore, the breathing system could also be exposed to these proteases, although to a lesser extent compared to the gastrointestinal tract, and could mainly represent an occupational occurrence.

Enzymes of PLCP family are involved in numerous physiological and pathological processes in different organisms, inside and outside the cells, since they can digest a wide range of substrates [6]. Their function includes the extracellular matrix remodelling, antigen presentation, hormone processing, parasite invasion and processing surface proteins [7,8]. Moreover, PLCP-mediated proteolysis is modulated by various parameters, including pH, ions and inhibitors. This flexible regulation contributes to maintain cell homeostasis and any disturbance of this network may lead to various disorders.

Some PLCPs can also cause allergic reactions [9], and those registered as allergens by the World Health Organisation and International Union of Immunological Societies (WHO/IUIS) Allergen Nomenclature Sub-Committee (https://allergen.org/), accessed on 2 January 2024, are listed in Table 1, where the code (EC number) given by the Enzyme Commission (EC) to the enzyme is also reported, when available.

In addition, Table 1 includes two representative proteases of the same family (ficin and papain), which have been described as sensitisers [9] because they are recognised and bound by specific IgE (http://www.allergome.org, accessed on 2 January 2024), but they are not (yet) registered by WHO/IUIS. It can be noted that the allergenic PLCPs officially recognized are three from plant foods (Act d 1, Ana c 2 and Cari p 2), one from pollen (Amb a 11) and six from mites (Blo t 1, Der f 1, Der m 1, Der p 1, Eur m 1 and Tyr p 1). Figure 2 shows that the exposure to PLCPs of mites and pollens mainly occurs by inhalation, thus meeting the respiratory system. Nevertheless, their ingestion can also occur, at least for a small number of molecules. For instance, we cannot exclude the ingestion of foods contaminated with dust containing mites or the ingestion of pollens, especially in periods of intense pollination.

Studies have pointed out that proteolytic activity associated with PLCPs may contribute to their allergenicity (Figure 3A), or to the allergenicity of other proteins, working like adjuvants [39]. This observation, together with the knowledge that plant and/or animal PLCPs can be found not only in natural sources but can be present in additional products, including foods, cosmetics, drugs and supplements, highlights the high relevance of these enzymes for industry and for health.

In this context, we have here analysed certain structural and functional details, substrate specificity, the regulation and inhibition of proteolytic activity, resistance to gastrointestinal digestion and allergenic evidence concerning PLCPs. A mention of specific arthropod and plant food proteases involved in allergic reactions and their sources is also reported. The contribution of their proteolytic activity to allergic reactions has been investigated.

## 2. PLCP Structural Features

Clan CA of PLCPs includes proteins with a papain-like fold [40,41,42]. These proteases are usually sensitive to the small molecule inhibitor E64 [43], which is ineffective against peptidases from other clans of cysteine peptidases. There are over thirty families in this clan, and tertiary structures have been solved for many members of it (as found in the PDB data bank, www.rcsb.org/, accessed on 2 January 2024) such as Act d 1 (2ACT, 1AEC), Cari p 2 (1YAL), Amb a 11 (5EGW, 5EF4), Der f 1 (5VPK), Der p 1 (1XKG, 2AS8, 3F5V) and papain (1KHP).

The PLCPs fold (Figure 3B) consists of two domains connected by a flexible linker with the active site between them [42]. One domain has a bundle of helices, with the catalytic Cys at the end of one of them, and the other subdomain is a β-barrel with the active site histidine (His) and asparagine (Asn). Asn is sometimes substituted by aspartate (Asp). In fact, the family of PLCPs is a classic example of enzymes requiring a cysteine residue as the catalytic nucleophile, and therefore, they are inhibited by thiol chelators such as iodoacetate, iodoacetic acid and *N*-ethylmaleimide orp-chloromercuribenzoate. In addition to the catalytic cysteine, the mechanism of action involves also a His and a Asn residue. All together, they constitute a generally conserved catalytic Cys–His–Asn triad lying at the surface of the cleft between the two domains of the molecule [8,41].

Although all known allergenic PLCPs are reported to show a similar fold, namely the papain-like fold [41], a comparative analysis of the primary structure of plant food enzymes and a comparison between plants and arthropods homologs highlights a low conservation of the protein sequence [9]. Conversely, a fairly high level of identity between the two mite allergens belonging to the Dermatophagoides genus, Der p 1 and Der f 1, is reported. In fact, the sequence identity observed between the analysed isoforms of Der p 1 and Der f 1 was about 82%, whereas the values obtained for the homologs from pineapple, papaya, kiwifruit and fig ranged from 29 to 59% [9]. In line with these structural features, the allergy reaction caused by the two mite PLCPs is almost overlapping, whereas the plant homologous proteins show individual immunological behaviours. All together, these features likely suggest a cross-reaction between Der p 1 and Der f 1 and the absence of cross-reactions between the mite PLCPs and plant food homologs. All these observations are in line with Chan et al. [44], reporting the WHO/IUIS observations suggesting that IgE cross-reactivity is possible at sequence identity values as low as 67%, whereas the probability decreases at lower values.

## 3. PLCP Substrates

PLCPs show proteolytic activity against a broad range of protein substrates (Table 1). The specificity of each protease towards its substrates is mostly defined by the structure of the active site. The papain active site has been widely studied for many years [45] and different features and models have been reported [46,47]. Figure 4 shows a drawing of one of the first models, proposed by Schechter and Berger [48] in 1970.

The scheme proposed in Figure 4 shows that the PLCP binding cleft can be divided into seven subsites, S1–S4 and S1′–S3′, each accommodating one amino acid residue of the peptidic substrate. The subsites are located on both sides of the catalytic site, four on the *N*-terminal side and three on the C-terminal side. Some years later, this model was revised, suggesting that the substrate residue binding regions beyond S2 and S2′ should not be called sites, but areas [49]. Despite this, after so many years, is still difficult to identify a general rule describing the active site and the substrate specificity of these proteases. What leaves no doubts is that different patterns of amino acid motif in this area affect the interaction between the protease and the ligands [6]. Unlike proteases such as the chymotrypsin-like class of serine proteases, which have a substrate specificity at the P1 position, PLCPs have been shown to have a primary substrate specificity at the P2 position [3,50]. Harris and colleagues [51] show that PLCPs usually show a preference for hydrophobic amino acids in the P2 position. Indeed, they observe that papain has a preference for P2 Val > Phe > Tyr, while it shows a preference for Pro in P3. The P4 position is reported to be very broad, but there is a lack of activity for large aliphatic and aromatic amino acids. It was also found [52] that the profile for the fastest substrates reveals that short aliphatic (Val, Ile, Leu, Ala) and that hydroxylic (Ser, Thr) amino acids are preferred by papain in the P2 position, whereas Gly, Asn and charged amino acids such as Asp, Glu, Arg, Lys and His are not well tolerated. In the P1 position, Lys, Gly and norleucine are favoured whereas Pro, His, Asp, Ile and Val are disfavoured. In the P3 position, papain has preference for Pro, Leu, Lys and Ile. Chruszcz and colleagues [42] show that two mite PLCPs, Der p 1 and Der f 1, prefer to bind small aliphatic residues in position P2, charged residues in position P1, and small hydrophobic or hydrophilic residues in position P1′. A summary of substrate specificity, as reported in [10], the enzyme nomenclature database (https://enzyme.expasy.org, accessed on 2 January 2024), is shown in Table 1.

An important example of PLCP substrates, the proteolysis of which affects the allergic response, are represented by the proteins of the tight junction complex (14), which have a regulatory function in the passage of ions and molecules through the paracellular pathway. In fact, the disruption of these molecules causes the transit of proteins, including the allergenic ones (Figure 5), through the tissue barriers, such as the lung and the gastrointestinal epithelia.

## 4. Inhibition of Enzyme Activity

Protein inhibitors, such as cystatins, can inhibit the enzyme activity of PLCPs. This inhibition is reported [53] to be competitive, noncovalent and reversible. Plant cystatines are also known as phytocystatins. These inhibitors are ubiquitous, and they are implicated in the regulation of both endogenous and exogenous proteases. In this way, they are involved in physiological and pathological processes, including defence from external agents. Cystatins can, therefore, also have an effect on the allergic reactions by regulating the activity of PLCPs.

E-64 is instead a non-protein inhibitor, which can irreversibly inhibit many PLCPs [54]. It is an epoxide, which was first isolated from Aspergillus japonicus in 1978 [55]. Since then, it has been widely used for experimental purposes, one of which is to investigate the function of PLCPs [56].

It is worth noting that the inhibition mechanism of PLCPs is still attracting researchers, and more recently, the allosteric regulation of activity, such as that of viral [57] and protozoan [58] proteases, was described. It would also be of interest to investigate the possible allosteric regulation of PLCPs from origins different from those already observed.

## 5. Resistance to Gastrointestinal Digestion

The effect of PLCPs on the gastrointestinal tissue depends on the resistance of the protease to gastrointestinal digestion. Actinidin has shown only a partial degradation after simulated gastrointestinal digestion [12,13], thus suggesting that it can exert the proteolytic activity on the met tissues of the gut. In contrast, other PLCPs, such as bromelain [19], ficin [33] and papain [37], are reported to be unstable to gastrointestinal digestion (Table 1). However, some PLCPs, such as bromelain, show very attracting therapeutic potentials, including analgesic, anti-inflammatory and anti-cancer activities [19]. To exploit these biological actions and overcome the difficulty of denaturation in the gastrointestinal environment, the pharmaceutical industry resorts to the oral delivery via encapsulation, which protects protein molecules, such as bromelain, avoiding its denaturation [48].

Data about the gastrointestinal resistance of the pollen and mite PLCPs are not available. In reality, in addition to food proteins, every protein known as inhaled allergen can also be ingested (Figure 2) and vice versa—some food allergenic proteins (Figure 1) can be inhaled [35,59,60]. For instance, it is possible for pollen allergens to be ingested in foods such as honey or in periods of pollination [61,62,63,64,65]. However, mite proteins, including the PLCP Der p 1 and its homologs, can be also ingested [28,66]. Therefore, the possibility that these PLCPs can be ingested makes the investigation of their stability on digestion highly relevant because their enzymatic activity could be involved in an adjuvant function with respect to allergenicity.

## 6. PLCP Allergenicity

PLCPs have been defined as initiator allergens promoting the development of allergic diseases by first impairing the epithelium, then recruiting immune cells and promoting the release of pro-inflammatory cytokines [67]. Their clinical relevance depends on the interface with protease inhibitors, the redox environment, their tendency to autolysis, the glycosylation pattern [68], their environmental distribution, human exposure and the levels and changes in the state of these allergens when in contact with the human immune system [67]. A different combination of these factors could explain why some food PLCPs, like Act d 1, which is partially resistant to gastrointestinal digestion, might be powerful allergens.

It is worth noting that the frequency of sensitisation to Der p 1 and to Der f 1, reported by Giangrieco et al. [9], was higher than that detected for plant food homologs in an Italian population, and that the sensitisation to food PLCPs was very rarely associated with clinical symptoms. Conversely, these two arthropod allergens, to which the exposure route is generally inhalation [69], provided an important contribution to the prevalence of symptoms common to mite allergy, such as rhinitis, bronchial asthma and conjunctivitis.

Some PLCPs may induce clinical reactions only in the presence of high levels of exposure to them, as occurs in some geographical areas [23] or in occupational diseases [70] or taking drugs with a high concentration of chymopapain [71] or bromelain [72]. In all other cases, it seems that some food PLCPs may induce the production of specific IgE in the absence of symptoms. In fact, IgE sensitisation and/or structural co-recognition between allergenic proteins and IgE antibodies is not necessarily associated with allergy reactions [73,74], as recently confirmed to also be the case in PLCPs [9]. A possible cross-reactivity between mite Der p 1 and F. carica ficin has been hypothesised [75], but no test has been performed to demonstrate the presence of specific IgE towards fig PLCP nor has any immunoinhibition test been carried out to demonstrate any possible cross-reactivity.

Literature reports have also indicated that proteolytic activity associated with PLCPs may contribute to their allergenicity (Figure 3A) or to the allergenicity of other proteins (Figure 5). In fact, this enzyme activity may work like an adjuvant, as reported for instance for Der p 1 [39]. This means that these proteases can act by enhancing the body’s immune response to an antigen. For instance, some PLCPs, although not evoking a specific IgE-mediated response, favour the penetration of allergens by increasing epithelial permeability initiating or exacerbating the allergic responses [76,77].

## 7. Sources of PLCPs Known for Allergenicity and/or IgE Binding

The list reported below of sources of PLCPs known for allergenicity and/or IgE binding is not comprehensive. Some possible sources, including soybean [78], kidney bean [79] and the mite Psoroptes ovis [80], are not described because the immunological characterisation of their PLCPs is not yet extensive. In addition, whether the soybean Gly m Bd 30K/P34 belongs to the family of PLCPs does not seem certain, since it is reported as “probable” cysteine protease of the papain family [78].

### 7.1. House Dust Mites (HDM)

HDM include several species of arthropods feeding on dead human skin cells and thrive in warm, humid environments. They belong to the order Astigmata with around 60 families. The most important are: Pyroglyphidae, Acaridae and Glycyphagidae. The Pyroglyphidae family dominates the domestic fauna with four species: *Dermatophagoides farinae* (American house dust mite), *Dermatophagoides pteronyssinus* (European house dust mite), *Euroglyphus maynei* (Mayne’s house dust mite) and *Dermatophagoides microceras*. HDM allergen exposure is a major risk factor for the development of persistent allergic respiratory diseases (Figure 2), such as asthma and allergic rhinoconjunctivitis, as well as symptoms such as atopic dermatitis [81].

Two related PLCPs, registered by the WHO/IUIS Allergen Nomenclature Sub-Committee as Der p 1 and Der f 1, were isolated from the mites *Dermatophagoides pteronyssinus* and *Dermatophagoides farinae*. They have been reported as major allergens present in the faeces of HDM [82]. Their allergenic properties are the most extensively studied and the correlation between sensitisation to Der p 1 and Der f 1 and clinical allergy to mites is often very frequent [9]. In addition to the best-known Der p 1 and Der f 1, four homologs from other mites have been registered by WHO/IUIS, namely Blo t 1, Der m 1, Eur m 1 and Tyr p 1, identified in *Blomia tropicalis*, *Dermatophagoides microceras*, *Euroglyphus maynei* and *Tyrophagus putrescentiae*, respectively (Table 1). It is also worth noting that Takai et al. [83] reported that the interaction of recombinant major mite group 1 allergens (Der f 1 and Der p 1) with an endogenous inhibitor, cystatin A ligand, may affect their allergenicity. In perspective, this observation appears very promising since it offers a possible therapeutic exploitation of PLCP inhibitors. In particular, these proteases are activated in an acidic environment, and they disrupt the tight junctions of human lung epithelium cells, causing increased transepithelial permeability [84] (Figure 5). In this context, a possible use of PLCP inhibitors could be set to prevent the tissue damage, which contributes to allergy reactions.

Animal models have shown an increase in eosinophil counts in the oesophagus after a nasal challenge with dust mites [85] and the sensitisation to Der p 1 and Der f 1 was associated with human eosinophilic oesophagitis [86]. Likewise eosinophilic oesophagitis has been described after desensitisation to dust mites with sublingual immunotherapy [87]. Moreover, dust mites can cause allergic symptoms if ingested raw [88]. Anyway, the mite group 1 allergens were suggested to be thermolabile [89], and therefore, they should not be held responsible of the allergic reaction that might occur through the ingestion of cooked foods contaminated by mites, as occurs in the pancake syndrome [89,90]. Further future studies are desirable in order to shed more light on this point.

Der p 1 triggers a proteolytic cascade in the digestive tract of the mite, activating the serine protease allergens Der p 3, Der p 6 and Der p 9. At the level of the airway epithelium, lung microbiome and secretome, it promotes the release of proinflammatory cytokines (IL-6, IL-8, GM-CSF, thymic stromal lymphopoietin and IL-25), alarmins (IL-1a and IL-33) and chemoattractants (CCL2 and CCL20), which activate dendritic cells (DC), basophils and ILC2s to promote the TH2 allergic response [91,92]. A similar effect was observed in the skin, where Der p 1 percutaneously led to inflammation [93], and in the human gut, where it directly affects gut function through its proteolytic activity [66].

Other homologous allergens belonging to the PLCP family which are present in house dust and foodstuffs have been registered by the WHO/IUIS (Table 1). Cross-reactivity and co-sensitisation between the allergens of Pyroglypidae mite species can be observed [94,95]. In tropical climates, the storage mite *Blomia tropicalis* is an important source of sensitising allergens. The storage mite *Tyrophagus putrescentiae* is a related mycophagous cheese mite associated with economic losses and/or health problems. It may be responsible for anaphylaxis through the ingestion of mite-contaminated foods and induces allergic respiratory symptoms after exposure [96,97,98]. On the other hand, some storage mites provide specific and desirable characteristics to certain traditional cheeses, such as the French Mimolette and the German Milbenkäse [99], but no allergic reactions have been assigned to them so far.

### 7.2. Green Kiwifruit

Green kiwifruit (*Actinidia deliciosa*) is the most common species of this fruit, which is consumed throughout the year in Italy, followed by *Actinidia chinensis*, the gold kiwifruit, available on the market for a few months. The gold kiwifruit shows some important differences when compared to the green species, such as a much lower expression level of the allergen actinidin [100,101]. Kiwifruit is most often consumed fresh, although it can be transformed into juice, purées and preserves, and is used as an ingredient in cooking.

An example of the several functions that can involve a PLCP is represented precisely by the kiwifruit actinidin enzyme. This is a PLCP found in very high amounts in kiwifruit, comprising up to 50% of soluble protein at harvest [102]. It was registered by the WHO/IUIS with the allergen name Act d 1 [103], (formerly named Act c 1, at the time when *Actinidia deliciosa* was named *Actinidia chinensis*). Act d 1 is a major kiwifruit allergen in monosensitised allergic patients [103], containing allergenic epitopes not only on its surface, since some of them are accessible to IgE just after thermal treatment [104]. In natural sources, the enzyme activity of actinidin modifies the concentration of the allergenic protein kiwellin, which can undergo proteolytic processing producing the polypeptide KiTH and the nutraceutical peptide kissper [101,105].

The permeabilisation action of a plant PLCP, namely actinidin, on intestinal cells and tissues has also been reported. The effect of this enzyme on intestinal cell monolayer integrity was investigated by Cavic et al. [14], who demonstrated that this protease exerts direct proteolytic cleavage on occludine (Table 1). The degradation of this structural protein increases the monolayer permeability and induces the passage of allergens. This observation was confirmed by Grozdanovic et al. [106] in experiments performed on a mouse model by measuring transepithelial resistance and in vivo intestinal permeability. The authors showed that the disruption of tight junctions by kiwifruit actinidin may increase intestinal permeability and contribute to the process of sensitisation in food allergies.

The proteolytic activity of this kiwifruit enzyme is exploited by the food industry (Figure 1), for instance as a meat tenderiser [107,108]. A further biotechnological application was described by Mostafaie et al. [15], who exploited the collagenolytic activity of actinidin to isolate different cell populations from various solid tissues, such as liver and thymus.

IgE-binding activity to actinidin was associated with severe (anaphylactic) reactions in some patient populations [109], while in other countries, actinidin was not reported as a major allergenic protein [110]. These variations might depend on the patterns of consumption [110], on different cultivars of kiwifruit containing different amounts of actinidin [100,102,111] and on the time of harvest influencing the level of expression [112]. Palacin and collaborators sustained a link between the amount of specific IgE to Act d 1 and anaphylaxis [109]. Some patients with kiwi allergy showed serum IgE reactivity to papain and bromelain [11,113]. A possible association between papain-induced occupational asthma and kiwifruit and fig allergy has been reported [114].

### 7.3. Papaya

Papaya (*Carica papaya* L., Caricaceae family) is a tropical plant, native of Central America. Its fruit can be eaten in both forms, unripe (as vegetable) and ripe (as fruit). Papaya pollen, latex and fruit are all sources of allergens.

The first cysteine protease isolated and characterised from *Carica papaya* was papain [115,116]. Three more major PLCPs (chymopapain, glycyl endopeptidase and caricain) have been identified and purified in papaya latex [117]. Chymopapain only has been registered by the WHO/IUIS as the allergen Cari p 2 [118], whereas papain (Cari p papain in Allergome database), caricain (Cari p caricain in Allergome database) and glycyl endopeptidase [119] are reported as IgE-binding proteins, suggesting that they are potential allergens. Nevertheless, the structure of chymopapain is extremely similar to those of the three other papaya proteinases. Differences in backbone conformation were found only for two loops at the surface of the chymopapain protein, far away from the active site cleft. At the same time, chymopapain is reported to be unique among the members of the papain family because it exists as a mixture of two forms, named A and B [120]. Reported jointly as EC 3.4.22.6, the two chymopapain forms are immunologically indistinguishable and have identical amino acid sequences, but they differ in their reactivity. Attempts to determine the structural basis for these observations were unsuccessful [120]. Chymopapain-based drugs, administered by intradiscal injection, were used in the past for the treatment of patients with herniated intervertebral discs (chemonucleolysis). However, this treatment is no longer used, as nearly 1% of the treated patients experienced anaphylaxis [71,119,121].

Among papaya PLCPs, papain is probably the one which has the most biotechnological applications and is more widely exploited at the industrial level, especially in the tenderising of meat products, in cheese-making and in the clarifying of beer (Figure 1). Moreover, it is an important reagent in the biochemical, immunochemical and pharmaceutical laboratory, as it has a wide range of bioactivities including the antioxidant, antibacterial and antiviral ones. In the textile industry, papain can be used for processing wool, boiling off cocoons and refining silks [122]. Therefore, similarly to other PLCPs, humans can be exposed to papain in numerous ways, not only directly eating papaya fruit but also eating foods treated with this protease or ingesting pharmaceuticals or through exposure to other sources. Papain is described to activate human mast cells to release proinflammatory mediators [123], and to activate the Transient Receptor Potential Vanilloid-type 1 (TRPV1+) sensory neurons directly, leading to Substance P release and to a feeling of itchiness [124]. Skin exposure to the cysteine protease papain promotes dendritic cell activation, mediating the degranulation of human eosinophils and the production of superoxide anion [125]. In geographical areas where papaya is intensively cultivated, its pollen, containing cysteine proteases, can cause respiratory symptoms followed by generalised reactions after the ingestion of papaya fruit [23]. Outside those geographical areas sensitisation to papaya does not usually occur from eating papaya fruit. In fact, rhinitis and asthma are reported to affect only workers of industries where papaya is handled [35,126,127,128]. A single case was reported many years ago [129] of a patient experiencing severe systemic allergic reaction after the ingestion of meat tenderiser.

### 7.4. Pineapple

Pineapple (*Ananas comosus*) is a tropical plant of the family Bromeliaceae, a native of South America, with an edible fruit. Its fruit is consumed fresh, cooked or extracted for its juice. Bromelain is a PLCP present in all parts of the pineapple plant, mainly extracted for the commercial use from the stem after the fruit has been harvested [130]. This protein has been registered by the WHO/IUIS with the allergen name Ana c 2 [18,131,132]. Some isoforms of Ana c 2 are glycosylated [9], and they are exploited to isolate MUXF3 [133], which is a protein fragment bearing the glycan moiety used as a marker for the detection of the IgEs specific for cross-reactive carbohydrate determinants (CCD). Anyway, only a fraction of the isoforms of this protease has an *N*-glycosylation site, which can bear a carbohydrate. The isoform officially recognised as an allergen by WHO/IUIS is Ana c 2.0101 (UniProt accession number O23791). It is noteworthy that the mature form of this isoform does not have the *N*-glycosylation site found in other isoforms [9].

Similar to other plant PLCPs, bromelain is exploited in many applications (Figure 1), such as in food, beverage, tenderisation, cosmetic, pharmaceutical and textile industries [130]. For example, this enzyme can be used in the baking industry because dough may be prepared more quickly if the gluten it contains has been partially hydrolysed [134]. In medicine, bromelain-based enzymatic debridement is an alternative to surgical eschar removal, and it is also employed as an anti-inflammatory drug. This enzyme is used for meat tenderising, but it is heat-labile and denatured in the cooking process. Bromelain has been implicated in allergic reactions after occupational exposure [70,135], and allergic sensitisation usually follows inhalation [131] and not ingestion [18]. Nevertheless, an anaphylactic reaction following the intake of an anti-inflammatory drug containing bromelain has been described [72].

### 7.5. Fig

The fig tree (*Ficus carica*) is native of the Mediterranean region, together with western and southern Asia. It belongs to the family of Moraceae and produces edible fruits, which are actually false fruits, also known as infructescence. Haesaerts et al. [136] reported that the fig latex contains a mixture of at least five cysteine proteases commonly known as ficins. They are present in different proportions that may change, depending on the health of the tree, the ambient conditions and watering. Moreover, it was reported that the content of ficins decreased during fruit ripening [137].

The fig infructescence is rarely reported as a cause of allergic reaction [138]. The “ficus-fruit syndrome” [139] was set up on the basis of the fact that some patients first appeared sensitised by the inhalation of airborne Ficus benjamina latex allergens and subsequently reacted to the fig fruit pulp and skin cross-reactive allergens [140]. PLCPs have been suggested as major allergens in this syndrome, as IgE against papain were found in some of these patients [60,139].

Fic c Ficin is a protein recognised by specific IgE. It is able to sensitise atopic subjects, but allergic reactions are attributable to a few case reports [9,60,114,138]. The cross-reactivity between papain and fig fruit has been documented by CAP inhibition [60].

Some authors have described ficins as non-glycosylated proteins [141]. However, as research progresses, new identified isoforms have been found glycosylated [142]. Moreover, PLCPs are multigene protein families, and therefore it is not surprising that many different isoforms are described by new studies.

Ficin is used (Figure 1) for the production of certain traditional cheeses [34,143]. In cheesemaking, the earliest references date back to the Iliad, written by Homer, who described fig-juice curdling milk in the seventh or the eighth century BCE. Moreover, it is also employed for the proteolysis of selected proteins, the production of bioactive peptides, milk clotting, meat tenderisation and the production of active antibody fragments [34,144].

### 7.6. Short Ragweed

Short ragweed (*Ambrosia artemisiifolia*) causes severe respiratory allergies in North America and Europe. The PLCP Amb a 11 is one of its major allergens, expressed as a combination of isoforms and glycoforms and recorded in the WHO/IUIS allergen database [25]. A new cysteine protease allergen from giant ragweed pollen (*Ambrosia trifida*), named as Amb t CP has been recently reported [145]. It has been associated with respiratory allergy in late summer and autumn.

## 8. Contribution to Allergenicity of Enzyme Activity and Tissue Damage

It has been shown that when PLCPs come into contact with a human organism, from whatever source they derive, they are able to degrade the tight junctions of the airway, intestine or skin tissues and thus allow allergens to enter (Figure 5), leading to sensitisation or, in already sensitised individuals, to an allergic response [76,146,147]. Therefore, the proteolytic activity of some PLCPs results substantial to cause sensitisation and allergic symptoms towards inhaled allergens [148] or towards food allergens stable to the gastrointestinal digestion [106]. Figure 6 shows a scheme of three possible events that might affect the contribution to allergenicity of PLCP enzyme activity and tissue damage.

The first event occurs when a working PLCP molecule reaches the epithelial tight junctions and disrupt them [14,59,84]. Then, following tissue permeabilisation, the proteases reach the mast cell, bind the exposed specific IgE and induce the release of inflammatory mediators, which activate the allergic reaction. This event can be triggered by inhaled PLCPs, which derive either from mites [66,148] or from plant foods [9,106], in the case of occupational exposures. However, the first event can also be induced by food PLCPs when they are resistant to gastrointestinal inactivation. In the second possible event, the allergic reaction is not triggered because PLCP enzyme activity is neutralized by a specific inhibitor [53,149]. In the third case, PLCP is denatured and/or fragmented in the gastrointestinal environment [19,33,37]. In both the second and third event, the protease is inactivated, thus preventing the tissue damage and the transit of molecules capable of inducing an allergic reaction.

## 9. Conclusions

The observation that allergic reactions to PLCPs are mostly detected for inhaled proteins suggest that gastrointestinal digestion could often neutralise these proteases, thus counteracting tissue damages. In contrast, the respiratory system does not have a similar protection and undergoes epithelial damage, which promotes permeabilisation and protein crossing through the tissue barrier. This mechanism might be involved in the sensitisation and allergic reactions towards mite and pollen PLCPs, and also in the occupational reactions towards plant food homologs, which can pass through the air and be inhaled. Clearly, on the basis of this mechanism, the allergic sensitisation and clinical reactions should be favoured by the tissue damage/permeabilisation. Therefore, the function of the enzyme protein, namely the proteolytic activity, could represent a factor that can favour the allergic reaction. In this case, the presence of a protease inhibitor could prevent this effect.

Due to the high sequence identity, it is understandable that there is co-recognition, associated with cross-reaction, between mite PLCPs. In contrast, the sequence identity among plant food homologs is low (see Table 2 reported by Giangrieco et al. [9]). However, despite the low sequence identity of their primary structure, the possibility of co-recognition among food PLCPs seems to exist based on the literature data reported in this review [11,113,114]. Nevertheless, although the positive results of IgE binding shown by in vitro tests [9] demonstrate the patients’ sensitisation, their association with clinical allergic reactions towards the PLCP sources is observed only rarely and generally occurs as occupational disease. Therefore, all these observations suggest that the food industry can exploit the function of plant PLCPs for products handling because, usually, these proteins do not represent a healthy risk, especially for consumers. Nevertheless, a certain level of attention will need to be paid to workers exposed to some sources who may suffer from occupational allergic reactions.

The available literature suggests that several factors can affect the sensitisation and clinical allergic reaction to PLCPs, including the level of exposure to these proteases, the exposed organs (skin, respiratory and gastrointestinal systems), enzyme sensitivity to gastrointestinal digestion, tissue damage and permeabilisation due to proteolytic activity, the simultaneous presence of other potential allergens and the presence of enzyme inhibitors. Therefore, what we observe as sensitisation and allergic reaction is derived from a complex combination of multiple factors. Further future studies will better clarify the contribution of each factor, including proteolytic activity, to sensitisation and allergic reactions to these proteases.

Thus, a better knowledge of PLCP function and regulation will be especially useful to manage patients suffering from reactions towards these allergenic molecules, especially the inhaled allergens. In fact, the detection and avoidance of airborne allergens is hardly feasible compared to ingested allergens. However, understanding the mechanisms and molecules involved will allow a better management and, hopefully, the neutralization of all PLCPs (both contained in foods and airborne) compromising our safety.

## Figures and Tables

**Figure 1 foods-13-00790-f001:**
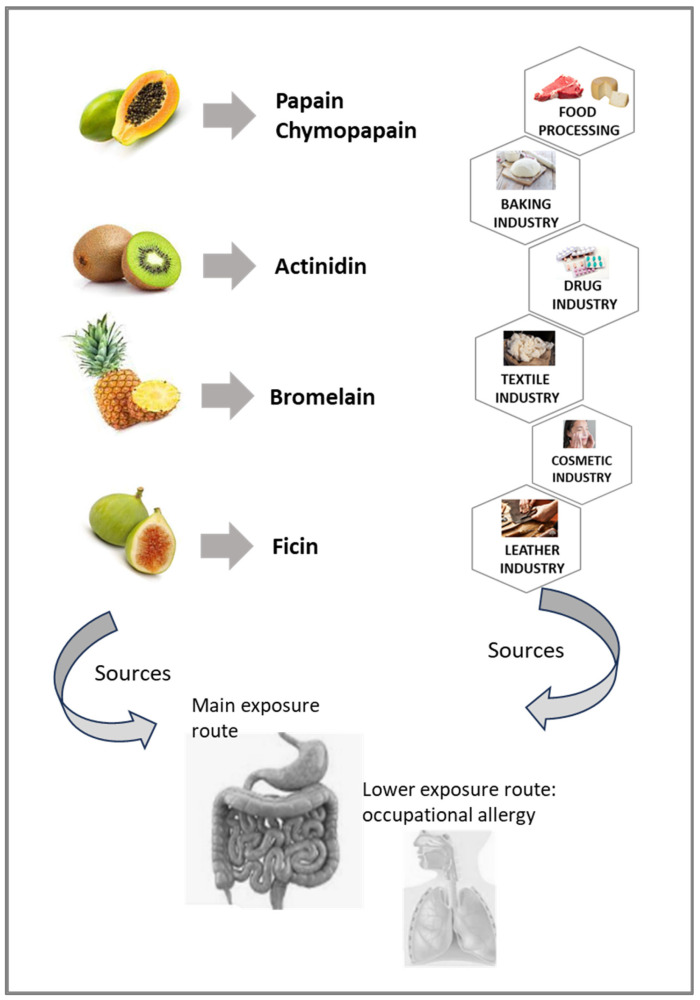
Sources of plant food PLCPs and exposure routes. The diagram shows the fruits (on the left) known as sources of the reported PLCPs (in between), which find several industrial applications (on the right).

**Figure 2 foods-13-00790-f002:**
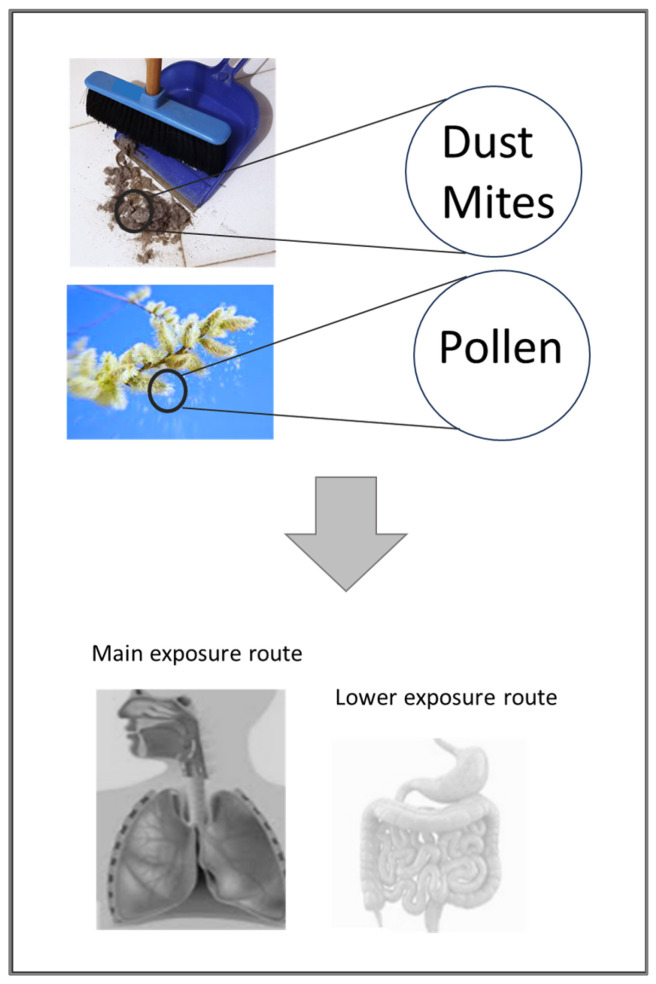
Diagram showing possible sources of commonly inhaled PLCPs and the exposure routes.

**Figure 3 foods-13-00790-f003:**
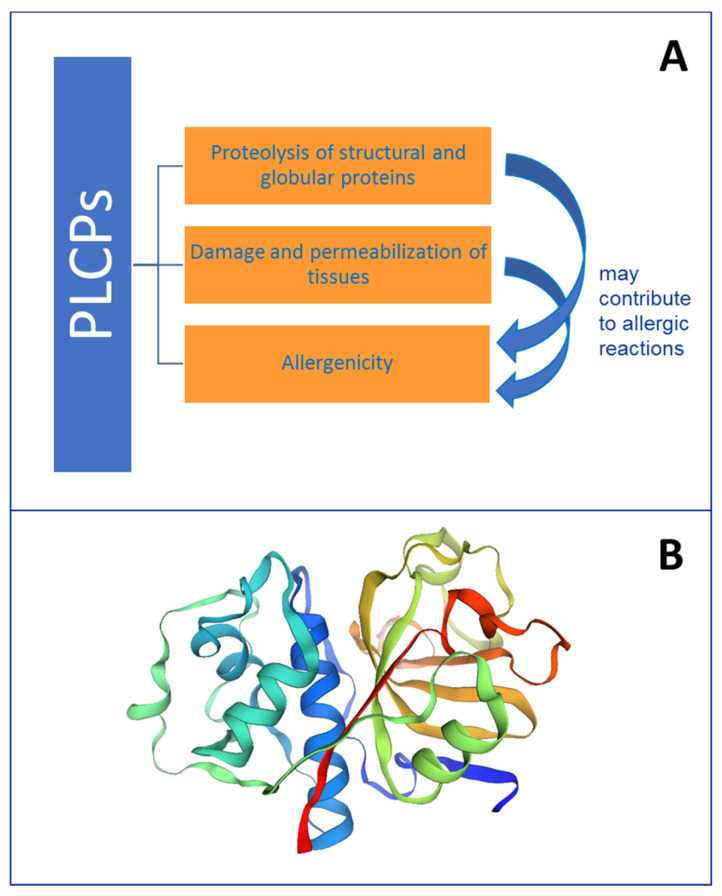
Schematic representation of some properties of PLCPs. Panel (**A**): summary of PLCP activities. Panel (**B**): 3D model of papain performed on the Expasy Swiss Bioinformatics Resource Portal (www.expasy.org/resources/swiss-model, accessed on 2 January 2024) by submitting the papain sequence with Uniprot code P00784.

**Figure 4 foods-13-00790-f004:**
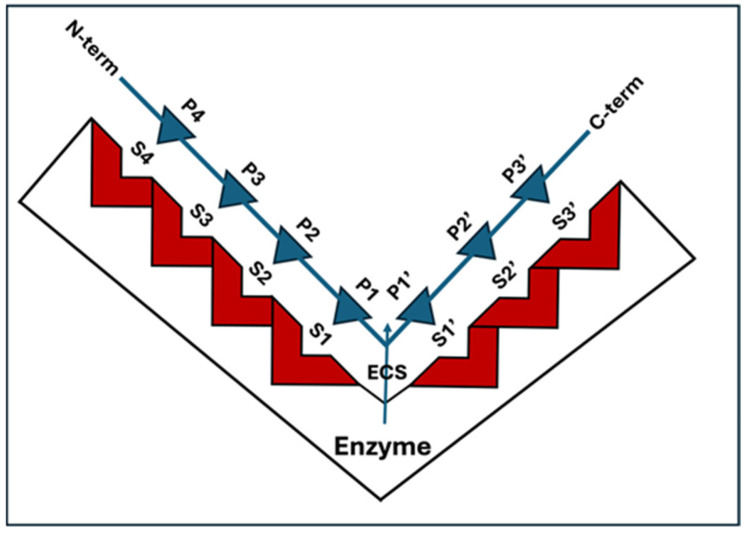
Schematic representation of a model of the complex PLCP substrate proposed by Schechter and Berger [48]. The binding cleft of the enzyme is composed of seven subsites, indicated as S1–S4 and S1′–S3′, located on both sides of the catalytic site. The enzyme cleavage site (ECS) is shown. The positions, P1–P4 and P1′–P3′, on the substrate are counted from the cleavage point with the same numbering as the subsites of the enzyme they occupy.

**Figure 5 foods-13-00790-f005:**
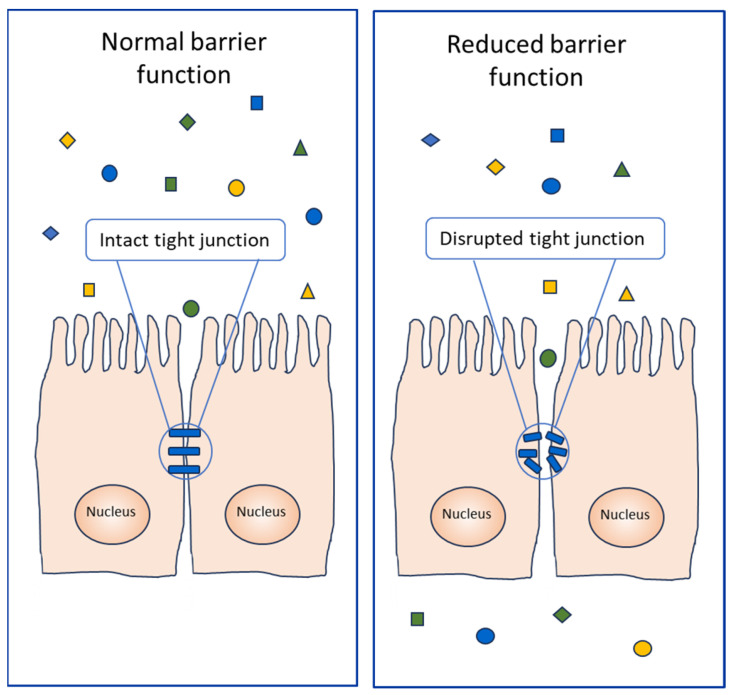
Schematic representation of the proteolytic activity of PLCPs on the epithelial tissues. Normal (on the (**left**)) and damaged (on the (**right**)) epithelia are shown. Protein molecules are represented by squares, triangles and rhombuses. Allergenic proteins are shown as circles.

**Figure 6 foods-13-00790-f006:**
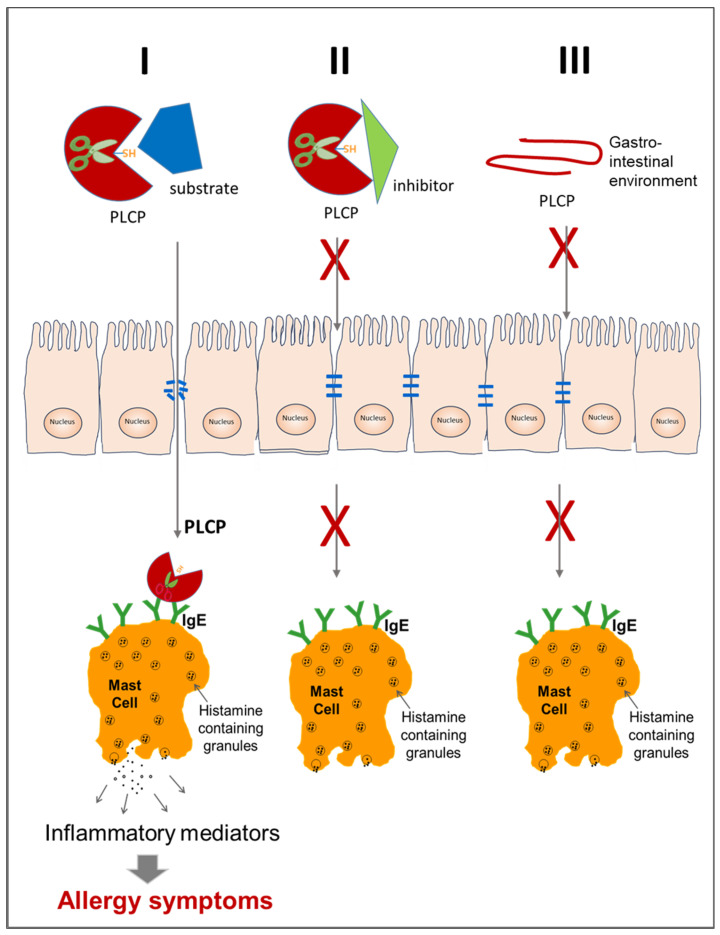
Schematic representation of three possible events (I, II and III) that can affect the contribution of PLCPs to allergenicity.

**Table 1 foods-13-00790-t001:** Characteristics of allergens belonging to the PLCP protein family found in the WHO-IUIS database, searched on 2 January 2024. At the bottom of Table 1, two representative potentially allergenic homologous proteins, not included in the WHO-IUIS database, are shown (the list is not comprehensive). Details can be found at WHO-IUIS website (http://allergen.org, accessed on 2 January 2024) or in the shown bibliography.

Species	Allergen	Biochemical Name/EC Number	Allergen Source	MW (SDS-PAGE)	Route of Exposure	Gastro-Intestinal Stability	Substrate (from Enzyme Nomenclature Database [10] or from the Indicated Bibliography) ^a^	Examples of Biotechnological Applications ^b^
ALLERGENIC PLCP FROM FOODS, POLLEN AND MITES								
*Actinidia deliciosa* (green kiwifruit)	Act d 1 [11]	Actinidin/EC 3.4.22.14	Food	30	Food	Stable [12,13]	Specificity close to that of papain. It disrupts the epithelial barrier of human intestinal T84 cells by degrading the tight junction protein occludin [14]. It hydrolyses collagen and fibrinogen [15,16].	Meat tenderisation [17]. Isolation of cells following collagenolytic digestion [15].
*Ananas comosus* (pineapple)	Ana c 2 [18]	Bromelain/EC 3.4.22.32	Food	22.8	Food, inhalation [18]	Unstable [19]	Hydrolysis of proteins with broad specificity for peptide bonds. Details can be found at www.enzyme-database.org/query.php?ec=3.4.22.32, accessed on 2 January 2024	Therapeutic and cosmetic use [17,20,21,22]. Meat tenderisation [17].
*Carica papaya* (papaya)	Cari p 2 [23]	Chymopapain/EC 3.4.22.6	Food	28	Food, inhalation [23]	ND	Specificity similar to that of papain	Allergenicity limits its application [24].
*Ambrosia artemisiifolia* (short ragweed)	Amb a 11 [25]	Cysteine protease	Pollen	37 kDa (mature protein), 52 kDa (zymogen)	Airway	ND	Not found	
*Blomia tropicalis* (storage mite)	Blo t 1 [26]	Cysteine protease	Mite	39	Airway	ND	Not found	
*Dermatophagoides farinae* (american house dust mite)	Der f 1 [27]	Cysteine protease/EC 3.4.22.65	Mite	27	Airway, ingestion [28]	ND	Not found	
*Dermatophagoides microceras* (house dust mite)	Der m 1 [29]	Cysteine protease	Mite	25	Airway	ND	Refers to Der p 1	
*Dermatophagoides pteronyssinus* (european house dust mite)	Der p 1 [30]	Cysteine protease/EC 3.4.22.65	Mite	24	Airway, ingestion [28]	ND	Details are available at www.enzyme-database.org/query.php?ec=3.4.22.65, accessed on 2 January 2024	
*Euroglyphus maynei* (house dust mite)	Eur m 1 [31]	Cysteine protease/EC 3.4.22.65	Mite		Airway	ND	Not found	
*Tyrophagus putrescentiae* (storage mite)	Tyr p 1 [32]	Cysteine protease	Mite	25	Airway	ND	Not found	
POTENTIALLY ALLERGENIC PLCP FROM FOODS, NOT (YET) REGISTERED BY WHO-IUIS								
*Ficus carica* (fig)	Fic c Ficin [9]	Ficin/EC 3.4.22.3	Food	24 [33]	Food, occupational	Unstable [33] need encapsulation	Specificity similar to that of papain	Meat tenderisation [17]. Cheese and milk protein hydrolysates for special foods production [34].
*Carica papaya* (papaya)	Cari p papain [9]	Papain/EC 3.4. 22.2	Food	24 [33]	Food, inhalation, skin contact [35,36]	Unstable [37] need encapsulation	Hydrolysis of proteins with broad specificity for peptide bonds, but preference for an amino acid bearing a large hydrophobic side chain at the P2 position. Does not accept Val in P1′.	Meat tenderization [17,38]. Exfoliantig agent [38].

^a^ Some details on substrates specificity are from Enzyme nomenclature database (https://enzyme.expasy.org, accessed on 2 January 2024). ^b^ These are examples only, the list is not comprehensive.

## Data Availability

No new data were created or analysed in this study. Data sharing is not applicable to this article.

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
