# Peer review of "Plant and Arthropod IgE-Binding Papain-like Cysteine Proteases: Multiple Contributions to Allergenicity"

_foods, 2024, doi:10.3390/foods13050790_

Round 1

Reviewer 1 Report

Comments and Suggestions for Authors

Kindly edit your manuscript using the feedback I provided in the attached file.

The key issue is that the abstract lacks the majority of your main findings, making it inferior to the body of the paper. Thus, I would like to summarise the most relevant results as described in the attached file, and I would appreciate your creativity in presentation.

Comments on the Quality of English Language

The English language requires too little revision.

Author Response

We thank very much the Reviewer for the revision of our manuscript and for all the comments and suggestions, which have been carefully considered and exploited to improve the manuscript. We hope that this revised version is now suitable for publication.

Please find below our point by point answers to the Reviewer’s comments.

REVIEWER 1. 

Comments and Suggestions for Authors

Kindly edit your manuscript using the feedback I provided in the attached file.

The key issue is that the abstract lacks the majority of your main findings, making it inferior to the body of the paper. Thus, I would like to summarise the most relevant results as described in the attached file, and I would appreciate your creativity in presentation.

ANSWERS TO THE REVIEWER COMMENTS

We thank the Reviewer for the postive comments and all the suggestions and the noted errors.

COMMENT 1

Line 88, “His and Asn”. Full AA name at 1st appearance

ANSWER 1

Full names of amino acids have been added and the text has been changed as follows: “…histidine (His) and asparagine (Asn). Asn is sometimes substitued by aspartate (Asp).”

COMMENT 2.

Line 103, “…the immunological behavior of…”. What reaction, allergy?

ANSWER 2

The text has been changed as follows: “the allergic reaction caused by…”

COMMENT 3

Line 109, delete “Enzyme”

ANSWER 3

The word “Enzyme” has been deleted

COMMENT 4

Lines 137-138. “However, some therapeutic potentials are very attracting, such as those of bromelain, that include analgesic, anti-inflammatory and anti-cancer activities [19].” Paraphrase to smoothly link next sentence.

ANSWER 4

To meet the Reviewer’s request, this sentence, and the next one, have been modified, as follows:

“However, some PLCPs, such as bromelain, show very attracting therapeutic potentials, including analgesic, anti-inflammatory and anti-cancer activities [19]. To exploit these biological actions, and overcome the difficulty of denaturation in the gastrointestinal environment, the pharmaceutical industry resorts to the oral delivery via encapsulation, which protects protein molecules, such as bromelain, avoiding its denaturation [48].”

COMMENT 5

Line 175, “hypotised”. Revise

ANSWER 5

The word has been changed to “hypothesized“

COMMENT 6

Line 180, “adjuvants”. Clarify.

ANSWER 6

This paragraph has been changed as follows:

“Literature reports have also indicated that proteolytic activity associated to PLCPs may contribute to their allergenicity (Figure 1A), or to the allergenicity of other proteins. In fact, this enzyme activity may working like an adjuvants, as reported for instance for Der p 1 [39]. This means that these proteases can act by enhancing the body's immune response to an antigen. For instance, some PLCPs, although not evoking a specific IgE-mediated response, favor the penetration of allergens by increasing epithelial permeability initiating or exacerbating the allergic responses [66,67].”

COMMENT 7

Line 222, ”.desensitisation”. ???

ANSWER 7

The mistake has been corrected by deleting the dot before the word "desensitisation".

COMMENT 8

Line 224, “termolabile”. Revise

ANSWER 8

The word has been changed to “thermolabile”

COMMENT 9

Line 231, “DCs”. ?????

ANSWER 9

The acronym has been specified with a full description as follows: “dendritic cells (DCs)”

COMMENT 10

Line 304, “Chymopain-“

ANSWER 10

The word has been changed to “Chymopapain-“

COMMENT 11

Line 318, TRPV1+. Full description

ANSWER 11

A full description has been included in the text.

COMMENT 12

Lines 321-325, and lines 395-404. You can use in abstract.

ANSWER

The suggested text has been deleted in the Section 6 and in the Conclusions, and it has been integrated in the abstract, taking into account that the abstract must not exceed 200 words.

COMMENT 13

Lines 416-418. It is the conclusion of your review, therefore summarise and highlight the most essential results or observations.

ANSWER 13

We thank the Reviewer for this suggestion. The text has been integrated to better summarise and highlight some results and observations.

COMMENT 14

Comments on the Quality of English Language

The English language requires too little revision.

ANSWER 14

English language has been checked

Reviewer 2 Report

Comments and Suggestions for Authors

The review article is good and pertains important information about the plant IgE-binding papain-like cysteine proteases.

1. The major concern regarding the article is its monotonous nature, which could be addressed by integrating data through the inclusion of Figures and Tables.

2. The author could provide illustrations of common PLCPs and define the regions associated with papain-like folds, among others. Additionally, describing the specific binding sites for inhibitors, whether they are competitive or allosteric, would be beneficial. 

3. A brief introduction, particularly in the pictorial form would be of great interest if author could make for the PLPC allergenicity focusing on the IgE-mediated molecular events.

4. Drawing from existing literature, propose some forward-looking research directions with a focus on food.

5. Some keywords redundantly mirror the title such as Papain-like cysteine proteases, allergy, IgE-binding, PLCPs; it is advisable to select keywords thoughtfully and avoid such duplication.

Comments on the Quality of English Language

Minor english editing

Author Response

We thank very much the Reviewer for the revision of our manuscript and for all the comments and suggestions, which have been carefully considered and exploited to improve the manuscript. We hope that this revised version is now suitable for publication.

Please find below our point by point answers to the Reviewer’s comments.

REVIEWER 2

Comments and Suggestions for Authors
The review article is good and pertains important information about the plant IgE-binding papain-like cysteine proteases.

ANSWERS TO THE REVIEWER COMMENTS

We thank the Reviewer for the positive comments and all the suggestions.

 COMMENT 1
1. The major concern regarding the article is its monotonous nature, which could be addressed by integrating data through the inclusion of Figures and Tables.

ANSWER 1

To address the Reviewer suggestion we have included 4 additional Figures. Now they are numbered as Figure 1, 2, 4 and 6. The text has been integrated accordingly.

COMMENT 2

  1. The author could provide illustrations of common PLCPs and define the regions associated with papain-like folds, among others. Additionally, describing the specific binding sites for inhibitors, whether they are competitive or allosteric, would be beneficial.

ANSWER 2

A paragraph (now paragraph 4) has been included to give details about the PLCP inhibitors and the type of inhibition. We also appreciate the suggestion to make a study of the regions associated with different papain-like folds. We consider this an aspect of great interest and for this reason we have planned to address it with an in-depth analysis in a future study.

 COMMENT 3

  1. A brief introduction, particularly in the pictorial form would be of great interest if author could make for the PLPC allergenicity focusing on the IgE-mediated molecular events.

ANSWER 3

As stated in the first answer, we have included new Figures to describe in a visual manner some events.

 COMMENT 4
4. Drawing from existing literature, propose some forward-looking research directions with a focus on food.
ANSWER 4

Some future perspectives have been integrated in the text, for instance at the end of the Conclusions paragraph and in the paragraph “6.1. House dust mites (HDM)”, now paragraph 7.1.

COMMENT 5

  1. Some keywords redundantly mirror the title such as Papain-like    cysteine proteases, allergy, IgE-binding, PLCPs; it is advisable to select keywords thoughtfully and avoid such duplication.
    ANSWER 5

Some keywords have been substituted

COMMENT 6

Comments on the Quality of English Language
Minor english editing

ANSWER 6

English editing has been carried out

Round 2

Reviewer 2 Report

Comments and Suggestions for Authors

The MS has been substantially modified. I recommend for the article publication.

Comments on the Quality of English Language

Minor English editing